# Modeling treatment events in disease progression

## Abstract

Ability to quantify and predict progression of a disease is fundamental for selecting an appropriate treatment. Many clinical metrics cannot be acquired frequently either because of their cost (e.g. MRI, gait analysis) or because they are inconvenient or harmful to a patient (e.g. biopsy, x-ray). In such scenarios, in order to estimate individual trajectories of disease progression, it is advantageous to leverage similarities between patients, i.e. the covariance of trajectories, and find a latent representation of progression. However, to our knowledge, most of existing methods for estimating trajectories do not explicitly account for treatment events such as surgery in-between observations, which dramatically decreases their adequacy for clinical practice. In this study, we develop a first machine learning framework named Coordinatewise-Soft-Impute (CSI) for analyzing disease progression from sparse observations in the presence of confounding events. CSI is easy to implement and is guaranteed to converge to the global minimum of the corresponding optimization problem. Experimental results also demonstrate the effectiveness of CSI using both simulated and real datasets.

## 1 Introduction

The course of disease progression in individual patients is one of the biggest uncertainties in medical practice. In an ideal world, accurate, continuous assessment of a patient's condition helps with prevention and treatment. However, many medical tests are either harmful or inconvenient to perform frequently, and practitioners have to infer the development of disease from sparse, noisy observations.

In its simplest form, the problem of modeling disease progressions is to fit the curve of $y(t), t \in [t_{\min}, t_{\max}]$ for each patient, given sparse observations $\mathbf{y} := (\tilde{y}(t_1), \ldots, \tilde{y}(t_n))$. Due to the high-dimensional nature of longitudinal data, existing results usually restrict solutions to subspace of functions and utilize similarities between patients via enforcing low-rank structures. One popular approach is the mixed effect models, including Gaussian process approaches (Verbeke, 1997; Zeger et al., 1988) and functional principal components (James et al., 2000). While generative models are commonly used and have nice theoretical properties, their result could be sensitive to the underlying distributional assumptions of observed data and hard to adapt to different applications. Another line of research is to pose the problem of disease progression estimation as an optimization problem. Kidzinski and Hastie. Kidziński & Hastie (2018) proposed a framework which formulates the problem as a matrix completion problem and solve it using matrix factorization techniques. This method is distribution-free and flexible to possible extensions.

Meanwhile, both types of solutions model the *natural* progression of disease using observations of the *targeted variables* only. They fail to incorporate the existence and effect of human interference: medications, therapies, surgeries, etc. Two patients with similar symptoms initially may have different futures if they choose different treatments. Without that information, predictions can be way-off.

To the best of our knowledge, existing literature talks little about modeling treatment effect on disease progression. In Kidziński & Hastie (2018), authors use concurrent observations of auxillary variables (e.g. oxygen consumption to motor functions) to help estimate the target one, under the assumption that both variables reflect the intrinsic latent feature of the disease and are thus correlated. Treatments of various types, however, rely on human decisions and to some extent, an exogenous variable to the development of disease. Thus they need to modeled differently.

In this work, we propose a model for tracking disease progression that includes the effects of treatments. We introduce the Coordinatewise-Soft-Impute (CSI) algorithm for fitting the model and investigate its theoretical and practical properties. The contribution of our work is threefold: First, we propose a model and an algorithm CSI, to estimate the progression of disease which incorporates the effect of treatment events. The framework is flexible, distribution-free, simple to implement and generalizable. Second, we prove that CSI converges to the global solution regardless of the initialization. Third, we compare the performance of CSI with various other existing methods on both simulated data and a dataset of Gillette Children's Hospital with patients diagnosed with Cerebral Palsy, and demonstrate the superior performances of CSI.

The rest of the paper is organized as follows. In Section 2 we state the problem and review existing methods. Next, in Section 3 we describe the model and the algorithm. Theoretic properties of the algorithm are derived in Section 4. Finally, in Section 5 and 6 we provides empirical results of CSI on the simulated and the real datesets respectively. We discuss some future directions in Section 7.

## 2 PROBLEM STATEMENT AND RELATED WORK

Let $y(t)$ be the trajectory of our objective variable, such as the size of tumor, over fixed time range $t \in [t_{\min}, t_{\max}]$, and $N$ be the number of patients. For each patient $1 \leq i \leq N$, we measure its trajectory $y_i(t)$ at $n_i$ irregularly time points $\mathbf{t}_i = [t_{i,1}, t_{i,2}, ..., t_{i,n_i}]'$ and denote the results as $\mathbf{y}_i = [y_{i,1}, ..., y_{i,n_i}]' = [y_i(t_{i,1}), ..., y_i(t_{i,n_i})]'$. We are primarily interested in estimating the disease progression trajectories $\{y_i(t)\}_{i=1}^N$ of all $N$ patients, based on observation data $\{(\mathbf{t}_i, \mathbf{y}_i)\}_{i=1}^N$.

To fit a continuous curve based on discrete observations, we restrict our estimations to a finite-dimensional space of functions. Let $\{b_i, i \in \mathbb{N}\}$ be a fixed basis of $L_2([t_{\min}, t_{\max}])$ (e.g. splines, Fourier basis) and $\mathbf{b} = \{b_i : 1 \leq i \leq K\}$ be first $K$ dimensions of it. The problem of estimating $y_i(t)$ can then be reduced to the problem of estimating the coefficients $\mathbf{w}_i = [w_{i,1}, w_{i,2}, \cdots, w_{i,K}]'$ such that $\mathbf{w}_i' \mathbf{b}(t)$ is close to $y_i(t)$ at time $t \in \mathbf{t}_i$.

Though intuitive, the above method has two main drawbacks. First, when the number of observations per patient is less than or equal to the number of basis functions $K$, we can perfectly fit any curve without error, leading to overfitting. Moreover, this direct approach ignores the similarities between curves. Different patients may share similar trend of the trajectories which could potentially imporve the prediction. Below we describe two main lines of research improving on this, the mixed-effect model and the matrix completion model.

### 2.1 LINEAR MIXED-EFFECT MODEL

In mixed-effect models, every trajectory $y_i(t)$ is assumed to be composed of two parts: the *fixed effect* $\mu(t) = \mathbf{m}'\mathbf{b}(t)$ for some $\mathbf{m} \in \mathbb{R}^K$ that remains the same among all patients and a *random effect* $\mathbf{w}_i \in \mathbb{R}^K$ that differs for each $i \in \{1, \ldots, N\}$. In its simplest form, we assume

$$\mathbf{w}_i \sim \mathcal{N}(\mathbf{0}, \Sigma) \quad \text{and} \quad \mathbf{y}_i | \mathbf{w}_i \sim \mathcal{N}(\mu_i + B_i \mathbf{w}_i, \sigma^2 \mathbb{I}_{n_i}),$$

where $\Sigma$ is the $K \times K$ covariance matrix, $\sigma$ is the standard deviation and $\mu_i = [\mu(t_{i,1}), \mu(t_{i,2}), \cdots \mu(t_{i,n_i})]'$, $B_i = [\mathbf{b}(t_{i,1}), \mathbf{b}(t_{i,2}), \cdots, \mathbf{b}(t_{i,n_i})]'$ are functions $\mu(t)$ and $\mathbf{b}(t)$ evaluated at the times $\mathbf{t}_i$, respectively. Estimations of model parameters $\mu, \Sigma$ can be made via expectation maximization (EM) algorithm (Laird & Ware, 1982). Individual coefficients $\mathbf{w}_i$ can be estimated using the best unbiased linear predictor (BLUP) (Henderson, 1975).

In linear mixed-effect model, each trajectory is estimated with $|\mathbf{w}_i| = K$ degrees of freedom, which can still be too complex when observations are sparse. One typical solution is to assume a low-rank structure of the covariance matrix $\Sigma$ by introducing a contraction mapping $A$ from the functional basis to a low-dimensional latent space. More specifically, one may rewrite the LMM model as

$$\mathbf{y}_i | \tilde{\mathbf{w}}_i \sim \mathcal{N}(\mu_i + B_i A \tilde{\mathbf{w}}_i, \sigma^2 \mathbb{I}_{n_i}),$$

where $A$ is a $K \times q$ matrix with $q < K$ and $\tilde{\mathbf{w}}_i \in \mathbb{R}^q$ is the new, shorter random effect to be estimated. Methods based on low-rank approximations are widely adopted and applied in practice and different algorithms on fitting the model have been proposed (James et al., 2000; Lawrence, 2004; Schulam & Arora, 2016). In the later sections, we will compare our algorithm with one specific implementation named functional-Principle-Component-Analysis (fPCA) (James et al., 2000), which uses EM algorithm for estimating model parameters and latent variables $\mathbf{w}_i$.

## 2.2 Matrix completion model

While the probabilistic approach of mixed-effect models offers many theoretical advantages including convergence rates and inference testing, it is often sensitive to the assumptions on distributions, some of which are hard to verify in practice. To avoid the potential bias of distributional assumptions in mixed-effect models, Kidzinski and Hastie (Kidziński & Hastie, 2018) formulate the problem as a sparse matrix completion problem. We will review this approach in the current section.

To reduce the continuous-time trajectories into matrices, we discretize the time range $[t_{\min}, t_{\max}]$ into $T$ equi-distributed points $G = [\tau_1, \ldots, \tau_T]$ with $\tau_1 = t_{\min}, \tau_T = t_{\max}$ and let $B = [\mathbf{b}(\tau_1), \mathbf{b}(\tau_2), \cdots, \mathbf{b}(\tau_T)]' \in \mathbb{R}^{T \times K}$ be the projection of the $K$-truncated basis $\mathbf{b}$ onto grid $G$. The $N \times K$ observation matrix $Y$ is constructed from the data $\{(\mathbf{t}_i, \mathbf{y}_i)\}_{i=1}^N$ by rounding the time $t_{i,j}$ of every observation $y_i(t_{i,j})$ to the nearest time grid and regarding all other entries as missing values. Due to sparsity, we assume that no two observation $y_i(t_{i,j})$'s are mapped to the same entry of $Y$.

Let $\Omega$ denote the set of all observed entries of $Y$. For any matrix $A$, let $P_\Omega(A)$ be the projection of $A$ onto $\Omega$, i.e. $P_\Omega(A) = M$ where $M_{i,j} = A_{i,j}$ for $(i,j) \in \Omega$ and $M_{i,j} = 0$ otherwise. Similarly, we define $P_\Omega^\perp(A) = A - P_\Omega(A)$ to be the projection on the complement of $\Omega$. Under this setting, the trajectory prediction problem is reduced to the problem of fitting a $N \times K$ matrix $W$ such that $WB' \approx Y$ on observed indices $\Omega$.

The direct way of estimating $W$ is to solve the optimization problem

$$\arg\min_W \frac{1}{2} \|P_\Omega(Y - WB')\|_F^2, \tag{2.1}$$

where $\| \cdot \|_F$ is the Fröbenius norm. Again, if $K$ is larger than the number of observations for some subject we will overfit. To avoid this problem we need some additional constraints on $W$. A typical approach in the matrix completion community is to introduce a nuclear norm penalty—a relaxed version of the rank penalty while preserving convexity (Rennie & Srebro, 2005; Candès & Recht, 2009). The optimization problem with the nuclear norm penalty takes form

$$\arg\min_W \frac{1}{2} \|P_\Omega(Y - WB')\|_F^2 + \lambda\|W\|_*, \tag{2.2}$$

where $\lambda > 0$ is the regularization parameter, $\| \cdot \|_F$ is the Fröbenius norm, and $\| \cdot \|_*$ is the nuclear norm, i.e. the sum of singular values. In Kidziński & Hastie (2018), a Soft-Longitudinal-Impute (SLI) algorithm is proposed to solve (2.2) efficiently. We refer the readers to Kidziński & Hastie (2018) for detailed description of SLI while noting that it is also a special case of our algorithm 1 defined in the next section with $\mu$ fixed to be 0.

## 3 Modeling treatment in disease progression

In this section, we introduce our model on effect of treatments in disease progression.

A wide variety of treatments with different effects and durations exist in medical practice and it is impossible to build a single model to encompass them all. In this study we take the simplified approach and regard treatment, with the example of one-time surgery in mind, as a non-recurring event with an additive effect on the targeted variable afterward. Due to the flexibility of formulation of optimization problem (2.1), we build our model based on matrix completion framework of Section 2.2.

More specifically, let $s(i) \in G$ be the time of treatment of the $i$'th patient, rounded to the closest $\tau_k \in G$ ($s(i) = \infty$ if no treatment is performed). We encode the treatment information as a $N \times T$ zero-one matrix $I_S$, where $(I_S)_{i,j} = 1$ if and only $\tau_j \geq s(i)$, i.e. patient $i$ has already taken the treatment by time $\tau_j$. Each row of $I_S$ takes the form of $(0, \cdots, 0, 1, \cdots, 1)$. Let $\mu$ denote the average additive effect of treatment among all patients. In practice, we have access to the sparse observation matrix $Y$ and surgery matrix $I_S$ and aim to estimate the treatment effect $\mu$ and individual coefficient matrix $W$ based on $Y, I_S$ and the fixed basis matrix $B$ such that $WB' + \mu I_S \approx Y$.

Again, to avoid overfitting and exploit the similarities between individuals, we add a penalty term on the nuclear norm of $W$. The optimization problem is thus expressed as:

$$\arg\min_{\mu,W} \frac{1}{2} \|P_\Omega(Y - WB' - \mu I_S)\|_F^2 + \lambda\|W\|_*, \tag{3.1}$$

for some $\lambda > 0$.

## 3.1 COORDINATEWISE-SOFT-IMPUTE (CSI) ALGORITHM

Though the optimization problem (3.1) above does not admit an explicit analytical solution, it is not hard to solve for one of $\mu$ or $W$ given the other one. For fixed $\mu$, the problem reduces to the optimization problem (2.2) with $\tilde{Y} = Y - \mu I_S$ and can be solved iteratively by the SLI algorithm Kidziński & Hastie (2018), which we will also specify later in Algorithm 1. For fixed $W$, we have

$$\arg\min_\mu \frac{1}{2}\|P_\Omega(Y - WB' - \mu I_S)\|_F^2 + \lambda\|W\|_*$$

$$= \arg\min_\mu \frac{1}{2}\|P_\Omega(-WB' - \mu I_S)\|_F^2 = \arg\min_\mu \frac{1}{2}\sum_{(i,j)\in\Omega\cap\Omega_S}((Y - WB')_{i,j} - \mu)^2, \quad (3.2)$$

where $\Omega_S$ is the set of non-zero indices of $I_S$. Optimization problem (3.2) can be solved by taking derivative with respect to $\mu$ directly, which yields

$$\hat{\mu} = \frac{\sum_{(i,j)\in\Omega\cap\Omega_S}(Y - WB')_{i,j}}{|\Omega \cap \Omega_S|}. \quad (3.3)$$

The clean formulation of (3.3) motivates us to the following Coordinatewise-Soft-Impute (CSI) algorithm (Algorithm 1): At each iteration, CSI updates $W_{\text{new}}$ from $(W_{\text{old}}, \mu_{\text{old}})$ via soft singular value thresholding and then updates $\mu_{\text{new}}$ from $(W_{\text{new}}, \mu_{\text{old}})$ via (3.3), finally it replaces the missing values of $Y$ based $(W_{\text{new}}, \mu_{\text{new}})$. In the definition, we define operator $S_\lambda$ as for any matrix $X$, $S_\lambda(X) := UD_\lambda V$, where $X = UDV$ is the SVD of $X$ and $D_\lambda = \text{diag}((\max\{d_i - \lambda, 0\})_{i=1}^K)$ is derived from the diagonal matrix $D = \text{diag}((d_i)_{i=1}^K)$. Note that if we set $\mu \equiv 0$ throughout the updates, then we get back to our base model SLI without treatment effect.

---

**Algorithm 1:** COORDINATEWISE-SOFT-IMPUTE

    1. Initialize $W_{\text{old}} \leftarrow$ all-zero matrix, $\mu_{\text{old}} \leftarrow 0$.

    2. Repeat:

        (a) Compute $W_{\text{new}} \leftarrow S_\lambda((P_\Omega(Y - \mu_{\text{old}}I_S) + P_\Omega^\perp(W_{\text{old}}B'))B)$;

        (b) Compute $\mu_{\text{new}} \leftarrow \frac{\sum_{(i,j)\in\Omega\cap\Omega_S}(Y - W_{\text{new}}B')_{i,j}}{|\Omega\cap\Omega_S|}$;

        (c) If $\max\left\{\frac{(\mu_{\text{new}} - \mu_{\text{old}})^2}{\mu_{\text{old}}^2}, \frac{\|W_{\text{new}} - W_{\text{old}}\|_F^2}{\|W_{\text{old}}\|_F^2}\right\} < \varepsilon$, exit;

        (d) Assign $W_{\text{old}} \leftarrow W_{\text{new}}, \mu_{\text{old}} \leftarrow \mu_{\text{new}}$.

    3. Output $\hat{W}_\lambda \leftarrow W_{\text{new}}, \hat{\mu}_\lambda \leftarrow \mu_{\text{new}}$.

---

## 4 CONVERGENCE ANALYSIS

In this section we study the convergence properties of Algorithm 1. Fix the regularization parameter $\lambda > 0$, let $(\mu_\lambda^{(k)}, W_\lambda^{(k)})$ be the value of $(\mu, W)$ in the $k$'th iteration of the algorithm, the exact definition of which is provided below in (4.4). We prove that Algorithm 1 reduces the loss function at each iteration and eventually converges to the global minimizer.

**Theorem 1.** *The sequence $(\mu_\lambda^{(k)}, W_\lambda^{(k)})$ converges to a limit point $(\hat{\mu}_\lambda, \hat{W}_\lambda)$ which solves the optimization problem:*

$$(\hat{\mu}_\lambda, \hat{W}_\lambda) = \arg\min_{\mu, W} \frac{1}{2}\|P_\Omega(Y - WB' - \mu I_S)\|_F^2 + \lambda\|W\|_*.$$

*Moreover, $(\hat{\mu}_\lambda, \hat{W}_\lambda)$ satisfies that*

$$\hat{W}_\lambda = S_\lambda((P_\Omega(Y - \hat{\mu}_\lambda I_S) + P_\Omega^\perp(\hat{W}_\lambda B'))B), \quad \hat{\mu}_\lambda = \frac{\sum_{(i,j)\in\Omega\cap\Omega_S}(Y - \hat{W}_\lambda B')_{i,j}}{|\Omega \cap \Omega_S|}. \quad (4.1)$$

The proof of Theorem 1 relies on five technique Lemmas stated below. The detailed proofs of the lemmas and the proof to Theorem 1 are provided in Appendix A. The first two lemmas are on properties of the nuclear norm shrinkage operator $S_\lambda$ defined in Section 3.1.

**Lemma 1.** *Let $W$ be an $N \times K$ matrix and $B$ is an orthogonal $T \times K$ matrix of rank $K$. The solution to the optimization problem $\min_W \frac{1}{2} \|Y - WB'\|_F^2 + \lambda \|W\|_*$ is given by $\hat{W} = S_\lambda(YB)$ where $S_\lambda(YB)$ is defined in Section 3.1.*

**Lemma 2.** *Operator $S_\lambda(\cdot)$ satisfies the following inequality for any two matrices $W_1$, $W_2$ with matching dimensions:*

$$\|S_\lambda(W_1) - S_\lambda(W_2)\|_F^2 \leq \|W_1 - W_2\|_F^2.$$

Define

$$f_\lambda(W, \mu) = \frac{1}{2} \|P_\Omega(Y - WB' - \mu I_S)\|_F^2 + \lambda \|W\|_*, \tag{4.2}$$

$$Q_\lambda(W|\tilde{W}, \mu) = \frac{1}{2} \|P_\Omega(Y - \mu I_S) + P_\Omega^\perp(\tilde{W}B') - WB'\|_F^2 + \lambda \|W\|_*. \tag{4.3}$$

Lemma 1 shows that in the $k$-th step of Algorithm 1, $W_\lambda^{(k)}$ is the minimizer for function $Q_\lambda(\cdot|W^{(k-1)}, \mu^{(k)})$. The next lemma proves the sequence of loss functions $f_\lambda(W_\lambda^{(k)}, \mu_\lambda^{(k)})$ is monotonically decreasing at each iteration.

**Lemma 3.** *For every fixed $\lambda \geq 0$, the $k$'th step of the algorithm $(\mu_\lambda^{(k)}, W_\lambda^{(k)})$ is given by*

$$W_\lambda^{(k)} = \arg\min_W Q_\lambda(W|W_\lambda^{(k-1)}, \mu_\lambda^{(k-1)}) \qquad \mu_\lambda^{(k)} = \frac{\sum_{(i,j)\in\Omega\cap\Omega_S}(Y - W_\lambda^{(k)}B')_{i,j}}{|\Omega \cap \Omega_S|}. \tag{4.4}$$

*Then with any starting point $(\mu_\lambda^{(0)}, W_\lambda^{(0)})$, the sequence $\{(\mu_\lambda^{(k)}, W_\lambda^{(k)})\}_k$ satisfies*

$$f_\lambda(W_\lambda^{(k)}, \mu_\lambda^{(k)}) \leq f_\lambda(W_\lambda^{(k)}, \mu_\lambda^{(k-1)}) \leq Q_\lambda(W_\lambda^{(k)}|W_\lambda^{(k-1)}, \mu_\lambda^{(k-1)}) \leq f_\lambda(W_\lambda^{(k-1)}, \mu_\lambda^{(k-1)}).$$

The next lemma proves that differences $(\mu_k - \mu_{k-1})^2$ and $\|W_\lambda^{(k)} - W_\lambda^{(k-1)}\|_F^2$ both converge to $0$.

**Lemma 4.** *For any positive integer $k$, we have $\|W_\lambda^{(k+1)} - W_\lambda^{(k)}\|_F^2 \leq \|W_\lambda^{(k)} - W_\lambda^{(k-1)}\|_F^2$. Moreover,*

$$\mu_\lambda^{(k+1)} - \mu_\lambda^{(k)} \to 0, \qquad W_\lambda^{(k+1)} - W_\lambda^{(k)} \to 0 \qquad as \qquad k \to \infty.$$

Finally we show that if the sequence $\{(\mu_\lambda^{(k)}, W_\lambda^{(k)})\}_k$, it has to converge to a solution of (4.1).

**Lemma 5.** *Any limit point $(\hat{\mu}_\lambda, \hat{W}_\lambda)$ of sequences $\{(\mu_\lambda^{(k)}, W_\lambda^{(k)})\}_k$ satisfies (4.1).*

## 5 SIMULATION STUDY

In this section we illustrate properties of our Coordinatewise-Soft-Impute (CSI) algorithm via simulation study. The simulated data are generated from a mixed-effect model with low-rank covariance structure on $W$:

$$Y = WB + \mu I_S + \mathcal{E},$$

for which the specific construction is deferred to Appendix B. Below we discuss the evaluation methods as well as the results from simulation study.

### 5.1 METHODS

We compare the Coordinatewise-Soft-Impute (CSI) algorithm specified in Algorithm 1 with the vanilla algorithm SLI (corresponding to $\hat{\mu} = 0$ in our notation) defined in Kidziński & Hastie (2018) and the fPCA algorithm defined in James et al. (2000) based on mixed-effect model. We train all three algorithms on the same set of basis functions and choose the tuning parameters $\lambda$ (for CSI and

SLI) and $R$ (for fPCA) using a 5-fold cross-validation. Each model is then re-trained using the whole training set and tested on a held-out test set $\Omega_{\text{test}}$ consisting 10% of all data.

The performance is evaluated in two aspects. First, for different combinations of the treatment effect $\mu$ and observation density $\rho$, we train each of the three algorithms on the simulated data set, and compute the relative squared error between the ground truth $\mu$ and estimation $\hat{\mu}$., i.e., $\text{RSE}(\hat{\mu}) = (\hat{\mu} - \mu)^2 / \mu^2$. Meanwhile, for different algorithms applied to the same data set, we compare the mean square error between observation $Y$ and estimation $\hat{Y}$ over test set $\Omega_{\text{test}}$, namely,

$$\text{MSE}(\hat{Y}) = \frac{1}{|\Omega_{\text{test}}|} \sum_{(i,j) \in \Omega_{\text{test}}} (Y_{ij} - \hat{Y}_{ij})^2 = \frac{1}{|\Omega_{\text{test}}|} \|P_{\Omega_{\text{test}}}(Y) - P_{\Omega_{\text{test}}}(\hat{Y})\|_F^2 \tag{5.1}$$

We train our algorithms with all combinations of treatment effect $\mu \in \{0, 0.2, 0.4, \cdots, 5\}$, observation rate $\rho \in \{0.1, 0.3, 0.5\}$, and thresholding parameter $\lambda \in \{0, 1, \cdots, 4\}$ (for CSI or SLI) or rank $R \in \{2, 3, \cdots, 6\}$ (for fPCA). For each fixed combination of parameters, we implemented each algorithm 10 times and average the test error.

## 5.2 RESULTS

The results are presented in Table 1 and Figure 1. From Table 1 and the left plot of Figure 1, we have the following findings:

1. CSI achieves better performance than SLI and fPCA, regardless of the treatment effect $\mu$ and observation rate $\rho$. Meanwhile SLI performs better than fPCA.

2. All three methods give comparable errors for smaller values of $\mu$. In particular, our introduction of treatment effect $\mu$ does not over-fit the model in the case of $\mu = 0$.

3. As the treatment effect $\mu$ increases, the performance of CSI remains the same whereas the performances of SLI and fPCA deteriorate rapidly. As a result, CSI outperforms SLI and fPCA by a significant margin for large values of $\mu$. For example, when $\rho = 0.1$, the $\text{MSE}(\hat{Y})$ of CSI decreases from 72.3% of SLI and 59.6% of fPCA at $\mu = 1$ to 12.4% of SLI and 5.8% of fPCA at $\mu = 5$.

4. All three algorithms suffer a higher $\text{MSE}(\hat{Y})$ with smaller observation rate $\rho$. The biggest decay comes from SLI with an average 118% increase in test error from $\rho = 0.5$ to $\rho = 0.1$. The performances of fPCA and CSI remains comparatively stable among different observation rate with a 6% and 12% increase respectively. This implies that our algorithm is tolerant to low observation rate.

To further investigate CSI's ability to estimate $\mu$, we plot the relative squared error of $\hat{\mu}$ using CSI with different observation rate in the right plot of Figure 1. As shown in Figure 1, regardless of the choice of observation rate $\rho$ and treatment effect $\mu$, $\text{RSE}(\hat{\mu})$ is always smaller than 1% and most of the estimations achieves error less than 0.1%. Therefore we could conclude that, even for sparse matrix $Y$, the CSI algorithm could still give very accurate estimate of the treatment effect $\mu$.

| Observation rate $\rho$ | | 0.1 | | | 0.3 | | | 0.5 | | |
|---|---|---|---|---|---|---|---|---|---|---|
| Treatment effect $\mu$ | | 1 | 2 | 5 | 1 | 2 | 5 | 1 | 2 | 5 |
| | fPCA | 0.521 | 2.172 | 5.455 | 0.525 | 2.039 | 5.170 | 0.525 | 2.036 | 5.166 |
| MSE($Y$) | SLI | 0.430 | 1.162 | 2.561 | 0.379 | 0.658 | 1.203 | 0.341 | 0.543 | 0.893 |
| | CSI | 0.311 | 0.306 | 0.318 | 0.314 | 0.297 | 0.320 | 0.294 | 0.299 | 0.295 |

Table 1: Comparisons between fPCA, SLI and CSI under different values of $\rho$ and $\mu$.

## 6 DATA STUDY

In this section, we apply our methods to real dataset on the progression of motor impairment and gait pathology among children with Cerebral Palsy (CP) and evaluate the effect of orthopaedic surgeries.

Cerebral palsy is a group of permanent movement disorders that appear in early childhood. Orthopaedic surgery plays a major role in minimizing gait impairments related to CP (McGinley et al.,

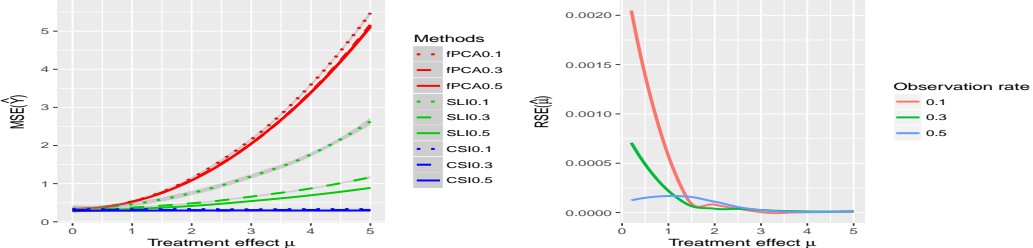

Figure 1: **Left: Comparisons between fPCA, SLI and CSI in estimating $Y$ with different observation rates.** Lines with colors red, green and blue correspond to fPCA, SLI and CSI respectively. Dotted, dashed and straight lines correspond to observation rate $\rho = 0.1, 0.3$ and $0.5$ respectively. **Right: Relationship between relative squared error of $\hat{\mu}$ and treatment effect $\mu$ using CSI with different observation rate.** Lines with colors red, green and blue correspond to observation rate $\rho = 0.1, 0.3$ and $0.5$ respectively.

2012). However, it could be hard to correctly evaluate the outcome of a surgery. For example, the seemingly positive outcome of a surgery may actually due to the natural improvement during puberty. Our objective is to single out the effect of surgeries from the natural progression of disease and use that extra piece of information for better predictions.

### 6.1 DATA AND METHOD

We analyze a data set of Gillette Children's Hospital patients, visiting the clinic between 1994 and 2014, age ranging between 4 and 19 years, mostly diagnosed with Cerebral Palsy. The data set contains 84 visits of 36 patients without gait disorders and 6066 visits of 2898 patients with gait pathologies. Gait Deviation Index (GDI), one of the most commonly adopted metrics for gait functionalities (Schwartz & Rozumalski, 2008), was measured and recorded at each clinic visit along with other data such as birthday, subtype of CP, date and type of previous surgery and other medical results.

Our main objective is to model individual disease progression quantified as GDI values. Due to insufficiency of data, we model surgeries of different types and multiple surgeries as a single additive effect on GDI measurements following the methodology from Section 3. We test the same three methods CSI, SLI and fPCA as in Section 5, and compare them to two benchmarks—the population mean of all patients (pMean) and the average GDI from previous visits of the same patient (rMean).

All three algorithms was trained on the spline basis of $K = 9$ dimensions evaluated at a grid of $T = 51$ points, with regularization parameters $\lambda \in \{20, 25, ..., 40\}$ for CSI and SLI and rank constraints $r \in \{2, \ldots, 6\}$ for fPCA. To ensure sufficient observations for training, we cross validate and test our models on patients with at least 4 visits and use the rest of the data as a common training set. The effective size of 2-fold validation sets and test set are 5% each. We compare the result of each method/combination of parameters using the mean square error of GDI estimations on held-out entries as defined in (5.1).

### 6.2 RESULTS

We run all five methods on the same training/validation/test set for 40 times and compare the mean and sd of test-errors. The results are presented in Table 2 and Figure 2. Compared with the null model pMean (Column 2 of Table 2), fPCA gives roughly the same order of error; CSI, SLI and rowMean provide better predictions, achieving 62%, 66% and 73% of the test errors respectively. In particular, our algorithm CSI improves the result of vanilla model SLI by 7%, it also provide a stable estimation with the smallest sd across multiple selections of test sets.

We take a closer look at the low-rank decomposition of disease progression curves provided by algorithms. Fix one run of algorithm CSI with $\lambda_\star = 30$, there are 6 non-zero singular value vectors, which we will refer as principal components. We illustrate the top 3 PCs scaled with corresponding singular values in Figure 3a. The first PC recovers the general trend that gait disorder develops

|        | mean   | scaled mean | sd    |
|-------:|-------:|------------:|------:|
| CSI    | 74.28  | 0.62        | 8.90  |
| SLI    | 79.92  | 0.66        | 9.22  |
| fPCA   | 127.73 | 1.06        | 13.54 |
| rMean  | 87.26  | 0.73        | 8.96  |
| pMean  | 119.80 | 1.00        | 12.84 |

Table 2: Test error on GDI dataset

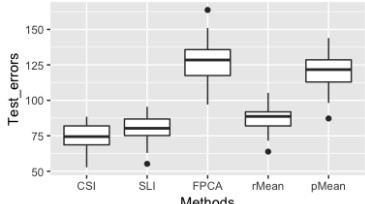

Figure 2: Box plot for test errors

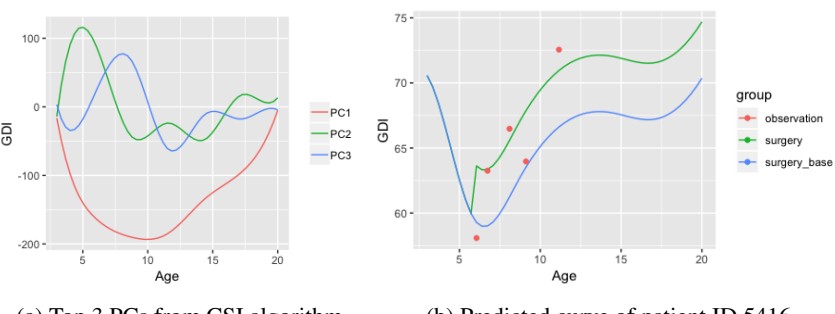

(a) Top 3 PCs from CSI algorithm

(b) Predicted curve of patient ID 5416

Figure 3: Low-rank decomposition of disease progression curves

through age 1-10 and partially recovers during puberty. The second and third PC reflects fluctuations during different periods of child growth. By visual inspection, similar trends can be find in the top components of SLI and fPCA as well.

An example of predicted curve from patient ID 5416 is illustrated in Figure 3b , where the blue curve represents the prediction without estimated treatment effect $\hat{\mu} = 4.33$, green curve the final prediction and red dots actual observations. It can be seen that the additive treatment effect helps to model the sharp difference between the exam before exam (first observation) and later exams.

## 7 CONCLUSION AND FUTURE WORK

In this paper, we propose a new framework in modeling the effect of treatment events in disease progression and prove a corresponding algorithm CSI. To the best of our knowledge, it's the first comprehensive model that explicitly incorporates the effect of treatment events. We would also like to mention that, although we focus on the case of disease progression in this paper, our framework is quite general and can be used to analyze data in any disciplines with sparse observations as well as external effects.

There are several potential extensions to our current framework. Firstly, our framework could be extended to more complicated settings. In our model, treatments have been characterized as the binary matrix $I_S$ with a single parameter $\mu$. In practice, each individual may take different types of surgeries for one or multiple times. Secondly, the treatment effect may be correlated with the latent variables of disease type, and can be estimated together with the random effect $w_i$. Finally, our framework could be used to evaluate the true effect of a surgery. A natural question is: does surgery really help? CSI provides estimate of the surgery effect $\mu$, it would be interesting to design certain statistical hypothesis testing/casual inference procedure to answer the proposed question.

Though we are convinced that our work will not be the last word in estimating the disease progression, we hope our idea is useful for further research and we hope the readers could help to take it further.

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

## A    PROOFS

*Proof of Lemma 1.* Note that the solution of the optimization problem

$$\min_A \frac{1}{2}\|Z - A\|_F^2 + \lambda\|A\|_* \tag{A.1}$$

is given by $\hat{A} = S_\lambda(Z)$ (see Cai et al. (2010) for a proof). Therefore it suffices to show the minimizer of the optimization problem (A.1) is the same as the minimizer of the following problem:

$$\min_W \frac{1}{2}\|YB - W\|_F^2 + \lambda\|W\|_*.$$

Using the fact that $\|A\|_F^2 = \text{Tr}(AA')$ and $B'B = \mathbb{I}_K$, we have

$$\arg\min_W \frac{1}{2}\|YB - W\|_F^2 + \lambda\|W\|_* = \arg\min_W \frac{1}{2}(\text{Tr}(YBB'Y') + \text{Tr}(WW') - 2\text{Tr}(YBW')) + \lambda\|W\|_*$$

$$= \arg\min_W \frac{1}{2}(\text{Tr}(WW') - 2\text{Tr}(YBW')) + \lambda\|W\|_*.$$

On the other hand

$$\arg\min_W \frac{1}{2}\|Y - WB'\|_F^2 + \lambda\|W\|_* = \arg\min_W \frac{1}{2}(\text{Tr}(YY') + \text{Tr}(WW') - 2\text{Tr}(YBW')) + \lambda\|W\|_*$$

$$= \arg\min_W \frac{1}{2}(\text{Tr}(WW') - 2\text{Tr}(YBW')) + \lambda\|W\|_*$$

$$= \arg\min_W \frac{1}{2}\|YB - W\|_F^2 + \lambda\|W\|_*$$

$$= S_\lambda(YB),$$

as desired. $\qquad\square$

*Proof of Lemma 2.* We refer the readers to the proof in Mazumder et al. (2010, Section 4, Lemma 3). $\qquad\square$

*Proof of Lemma 3.* First we argue that $\mu_\lambda^{(k)} = \arg\min_\mu f_\lambda(W_\lambda^{(k)}, \mu)$ and the first inequality immediately follows. We have

$$\arg\min_\mu f_\lambda(W_\lambda^{(k)}, \mu) = \arg\min_\mu \|P_\Omega(Y - W_\lambda^{(k)}B' - \mu I_S)\|_F^2$$

$$= \arg\min_\mu \sum_{(i,j)\in\Omega\cap\Omega_S} ((Y - W_\lambda^{(k)}B')_{i,j} - \mu)^2.$$

Taking derivative with respect to $\mu$ directly gives $\mu_\lambda^{(k)} = \arg\min_\mu f_\lambda(W_\lambda^{(k)}, \mu)$, as desired.

For the rest two inequalities, notice that

$$f_\lambda(W_\lambda^{(k)}, \mu_\lambda^{(k-1)}) = \frac{1}{2}\|P_\Omega(Y - W_\lambda^{(k)}B' - \mu_\lambda^{(k-1)}I_S)\|_F^2 + \lambda\|W_\lambda^{(k)}\|_*$$

$$\leq \frac{1}{2}\|P_\Omega(Y - \mu_\lambda^{(k-1)}I_S) + P_\Omega^\perp(W_\lambda^{(k-1)}B') - W_\lambda^{(k)}B'\|_F^2 + \lambda\|W_\lambda^{(k)}\|_* \tag{A.2}$$

$$= Q_\lambda(W_\lambda^{(k)}|W_\lambda^{(k-1)}, \mu_\lambda^{(k-1)})$$

$$\leq Q_\lambda(W_\lambda^{(k-1)}|W_\lambda^{(k-1)}, \mu_\lambda^{(k-1)}) \tag{A.3}$$

$$= \frac{1}{2}\|P_\Omega(Y - W_\lambda^{(k-1)}B' - \mu_\lambda^{(k-1)}I_S)\|_F^2 + \lambda\|W_\lambda^{(k-1)}\|_*$$

$$= f_\lambda(W_\lambda^{(k-1)}, \mu_\lambda^{(k-1)}).$$

Here the (A.2) holds because we have

$$\frac{1}{2}\|P_\Omega(Y - \mu_\lambda^{(k-1)}I_S) + P_\Omega^\perp(W_\lambda^{(k-1)}B') - W_\lambda^{(k)}B'\|_F^2$$

$$=\frac{1}{2}\|P_\Omega(Y - \mu_\lambda^{(k-1)}I_S - W_\lambda^{(k)}B') + P_\Omega^\perp(W_\lambda^{(k-1)}B' - W_\lambda^{(k)}B')\|_F^2$$

$$=\frac{1}{2}\|P_\Omega(Y - \mu_\lambda^{(k-1)}I_S - W_\lambda^{(k)}B')\|_F^2 + \frac{1}{2}\|P_\Omega^\perp(W_\lambda^{(k-1)}B' - W_\lambda^{(k)}B')\|_F^2$$

$$\geq\frac{1}{2}\|P_\Omega(Y - \mu_\lambda^{(k-1)}I_S - W_\lambda^{(k)}B')\|_F^2.$$

(A.3) follows from the fact that $W_\lambda^{(k)} = \arg\min_W Q_\lambda(W|W_\lambda^{(k-1)}, \mu_\lambda^{(k-1)})$. $\qquad\square$

*Proof of Lemma 4.* First we analyze the behavior of $\{\mu_\lambda^{(k)}\}$,

$$f_\lambda(W_\lambda^{(k)}, \mu_\lambda^{(k-1)}) - f_\lambda(W_\lambda^{(k)}, \mu_\lambda^{(k)}) = \frac{1}{2}\|P_\Omega(Y - W_\lambda^{(k)}B' - \mu_\lambda^{(k-1)}I_S)\|_F^2$$

$$-\frac{1}{2}\|P_\Omega(Y - W_\lambda^{(k)}B' - \mu_\lambda^{(k)}I_S)\|_F^2$$

$$=\frac{|S \cap \Omega_S|}{2}(\mu_\lambda^{(k)} - \mu_\lambda^{(k-1)})^2.$$

Meanwhile, the sequence $(\cdots, f_\lambda(W_\lambda^{(k-1)}, \mu_\lambda^{(k-1)}), f_\lambda(W_\lambda^{(k)}, \mu_\lambda^{(k-1)}), f_\lambda(W_\lambda^{(k)}, \mu_\lambda^{(k)}), \cdots)$ is decreasing and lower bounded by $0$ and therefore converge to a non-negative number, yielding the differences $f_\lambda(W_\lambda^{(k)}, \mu_\lambda^{(k-1)}) - f_\lambda(W_\lambda^{(k)}, \mu_\lambda^{(k)}) \to 0$ as $k \to \infty$. Hence

$$\mu_\lambda^{(k)} - \mu_\lambda^{(k-1)} \to 0, \tag{A.4}$$

as desired.

The sequence $\{W_\lambda^{(k)}\}$ is slightly more complicated, direct calculation gives

$$\|W_\lambda^{(k)} - W_\lambda^{(k-1)}\|_F^2 = \|S_\lambda(P_\Omega(Y - \mu_\lambda^{(k-1)}I_S) + P_\Omega^\perp(W_\lambda^{(k-1)}B'))$$

$$- S_\lambda(P_\Omega(Y - \mu_\lambda^{(k-2)}I_S) + P_\Omega^\perp(W_\lambda^{(k-2)}B'))\|_F^2$$

$$\leq \|P_\Omega(Y - \mu_\lambda^{(k-1)}I_S) + P_\Omega^\perp(W_\lambda^{(k-1)}B') \tag{A.5}$$

$$- P_\Omega(Y - \mu_\lambda^{(k-2)}I_S) - P_\Omega^\perp(W_\lambda^{(k-2)}B')\|_F^2$$

$$= |\Omega \cap \Omega_S|(\mu_\lambda^{(k-1)} - \mu_\lambda^{(k-2)})^2 + \|P_\Omega^\perp(W_\lambda^{(k-1)}B' - W_\lambda^{(k-2)}B')\|_F^2, \tag{A.6}$$

where (A.5) follows from Lemma 2, (A.6) can be derived pairing the 4 terms according to $P_\Omega$ and $P_\Omega^\perp$.

By definition of $\mu_\lambda^{(k)}$, we have

$$|\Omega \cap \Omega_S|(\mu_\lambda^{(k-1)} - \mu_\lambda^{(k-2)})^2 = \frac{1}{|\Omega \cap \Omega_S|}\left(\sum_{(i,j)\in\Omega\cap\Omega_S}(W_\lambda^{(k-1)}B' - W_\lambda^{(k-2)}B')_{i,j}\right)^2$$

$$\leq \|P_\Omega(W_\lambda^{(k-1)}B' - W_\lambda^{(k-2)}B')\|_F^2, \tag{A.7}$$

where (A.7) follows from the Cauchy-Schwartz inequality.

Combining (A.6) with (A.7), we get

$$\|W_\lambda^{(k)} - W_\lambda^{(k-1)}\|_F^2 \leq \|W_\lambda^{(k-1)}B' - W_\lambda^{(k-2)}B'\|_F^2 = \|W_\lambda^{(k-1)} - W_\lambda^{(k-2)}\|_F^2.$$

Now we are left to prove that the difference sequence $\{W_\lambda^{(k)} - W_\lambda^{(k-1)}\}$ converges to zero. Combining (A.4) and (A.7) it suffices to prove that $\|P_\Omega^\perp(W_\lambda^{(k-1)}B' - W_\lambda^{(k-2)}B')\|_F^2 \to 0$. We have

$$f_\lambda(W_\lambda^{(k-1)}, \mu_\lambda^{(k-2)}) - Q_\lambda(W_\lambda^{(k-1)}|W_\lambda^{(k-2)}, \mu_\lambda^{(k-2)}) = -\|P_\Omega^\perp(W_\lambda^{(k-1)}B' - W_\lambda^{(k-2)}B')\|_F^2,$$

and the left hand side converges to 0 because

$$0 \geq f_\lambda(W_\lambda^{(k-1)}, \mu_\lambda^{(k-2)}) - Q_\lambda(W_\lambda^{(k-1)}|W_\lambda^{(k-2)}, \mu_\lambda^{(k-2)})$$
$$\geq f_\lambda(W_\lambda^{(k-2)}, \mu_\lambda^{(k-2)}) - f_\lambda(W_\lambda^{(k-1)}, \mu_\lambda^{(k-2)}) \to 0,$$

which completes the proof. □

*Proof of Lemma 5.* Let $(\mu_\lambda^{(m_k)}, W_\lambda^{(m_k)}) \to (\hat{\mu}_\lambda, \hat{W}_\lambda)$, then Lemma 4 gives $(\mu_\lambda^{(m_k-1)}, W_\lambda^{(m_k-1)}) \to (\hat{\mu}_\lambda, \hat{W}_\lambda)$. Since we have

$$W_\lambda^{(m_k)} = S_\lambda((P_\Omega(Y - \mu_\lambda^{(m_k-1)} I_S) + P_\Omega^\perp(W_\lambda^{(m_k-1)} B'))B),$$
$$\mu_\lambda^{(m_k)} = \frac{\sum_{(i,j)\in\Omega\cap\Omega_S}(Y - W_\lambda^{(m_k-1)} B')_{i,j}}{|\Omega \cap \Omega_S|}.$$

Taking limits on both sides gives us the desire result. □

*Proof of Theorem 1.* Let $(\hat{\mu}_\lambda, \hat{W}_\lambda)$ be one limit point then we have:

$$\|\hat{W}_\lambda - W_\lambda^{(k)}\|_F = \|S_\lambda((P_\Omega(Y - \hat{\mu}_\lambda I_S) + P_\Omega^\perp(W_\lambda B'))B) \qquad \text{(A.8)}$$
$$- S_\lambda((P_\Omega(Y - \hat{\mu}_\lambda^{(k-1)} I_S) + P_\Omega^\perp(W_\lambda^{(k-1)} B'))B)\|_F^2$$
$$\leq |\Omega \cap \Omega_S|(\hat{\mu}_\lambda - \mu_\lambda^{(k-1)})^2 + \|P_\Omega^\perp((\hat{W}_\lambda - W_\lambda^{(k-1)})B')\|_F^2, \qquad \text{(A.9)}$$

here (A.8) uses Lemma 5 and (A.9) uses Lemma 2. Meanwhile,

$$|\Omega \cap \Omega_S|(\hat{\mu}_\lambda - \mu_\lambda^{(k-1)})^2 = \frac{\sum_{(i,j)\in\Omega\cap\Omega_S}((\hat{W}_\lambda - W_\lambda^{(k-1)})B')_{i,j}^2}{|\Omega \cap \Omega_S|} \leq \|P_\Omega((\hat{W}_\lambda - W_\lambda^{(k-1)})B')\|_F^2.$$
$$\text{(A.10)}$$

Combining (A.9) and (A.10), we have

$$\|\hat{W}_\lambda - W_\lambda^{(k)}\|_F \leq \|\hat{W}_\lambda - W_\lambda^{(k-1)}\|_F.$$

Hence the sequence $\|\hat{W}_\lambda - W_\lambda^{(k)}\|_F$ is monotonically decreasing and has a limit. But since there exists $W_\lambda^{(m_k)}$ converging to $\hat{W}_\lambda$, the limit equals 0, which proves $W_\lambda^{(k)} \to \hat{W}_\lambda$, $\mu_\lambda^{(k)} \to \hat{\mu}$.

Therefore we have proved the sequence $(\mu_\lambda^{(k)}, W_\lambda^{(k)})$ always converges.

Meanwhile, the first part of equation 4.1 and Lemma 5 in Mazumder et al. (2010) guarantees $\vec{0} \in \partial_W f_\lambda(\hat{W}_\lambda, \hat{\mu})$. By taking derivative directly we have $0 = \partial_\mu f_\lambda(\hat{W}_\lambda, \hat{\mu})$. Therefore $(\hat{W}_\lambda, \hat{\mu})$ is a stationary point for $f_\lambda(W, \mu)$. Notice that the loss function $f_\lambda(W, \mu)$ is a convex function with respect to $(W, \mu)$. Thus we have proved that the limit point $(\hat{W}_\lambda, \hat{\mu})$ minimizes the function $f_\lambda(W, \mu)$. □

## B  DATA GENERATION

Let $G$ be the grid of $T$ equidistributed points and let $B$ be the basis of $K$ spline functions evaluated on grid $G$. We will simulate the $N \times K$ observation matrix $Y$ with three parts

$$Y = WB + \mu I_S + \mathcal{E},$$

where $W$ follows a mixture-Gaussian distribution with low rank structure, $I_S$ is the treatment matrix with uniformly distributed starting time and $\mathcal{E}$ represents the i.i.d. measurement error. The specific procedures is described below.

1. Generating $W$ given parameters $\kappa \in (0, 1), r_1, r_2 \in \mathbb{R}, s_1, s_2 \in \mathbb{R}_{\geq 0}^K$:
    (a) Sample two $K \times K$ orthogonal matrices $V_1, V_2$ via singular-value-decomposing two random matrix.
    (b) Sample two unit length $K$ vectors $\vec{\gamma}_1, \vec{\gamma}_2$ via normalizing i.i.d. normal samples.

(c) Draw vector $\vec{t} \in \mathbb{R}^N$ from i.i.d. Bernoulli($\kappa$) samples. Denote the all one vector by $\vec{1}$.

(d) Draw $N \times K$ matrices $U_1, U_2$ from i.i.d. standard normal random variables.

(e) Set

$$W \leftarrow \vec{t} \cdot \left[ r_1 \vec{\gamma}_1 + U_1 \operatorname{diag}[\sqrt{s_1}] V_1 \right] + (\vec{1} - \vec{t}) \cdot \left[ r_2 \vec{\gamma}_2 + U_2 \operatorname{diag}[\sqrt{s_2}] V_2 \right],$$

where $\operatorname{diag}[s]$ is the diagonal matrix with diagonal elements $s$, "$\cdot$" represents coordinatewise multiplication, and we are recycling $\vec{t}, \vec{1} - \vec{t}$ and $r_i \vec{\gamma}_i$ to match the dimension.

2. Generating $I_S$ given parameter $p_{\text{tr}} \in (0, 1)$.

(a) For each $k = 1, \ldots, N$, sample $T_k$ uniformly at random from $\{1, \ldots, \lfloor T/p_{\text{tr}} \rfloor\}$.

(b) Set $I_S \leftarrow (\mathbf{1}\{j \geq T_i\})_{1 \leq i \leq N, 1 \leq j \leq T}$.

3. Given parameter $\epsilon \in \mathbb{R}_{\geq 0}$, $\mathcal{E}$ is drawn from from i.i.d. Normal$(0, \epsilon^2)$ samples.

4. Given parameter $\mu \in \mathbb{R}$, let $Y_0 \leftarrow WB + \mu I_S + \mathcal{E}$.

5. Given parameter $\rho \in (0, 1)$, drawn 0-1 matrix $I_\Omega$ from i.i.d. Bernoulli($\rho$) samples. Let $\Omega$ denote the set of non-zero entries of $I_\Omega$, namely, the set of observed data. Set

$$Y \leftarrow (Y_{ij})_{1 \leq i \leq N, 1 \leq j \leq T}, \quad \text{where } Y_{ij} = \begin{cases} (Y_0)_{ij} & \text{if } (I_\Omega)_{ij} = 1 \\ \text{NA} & \text{otherwise} \end{cases}.$$

In actual simulation, we fix the auxiliary parameters as follows,

$$K = 7, T = 51, N = 500,$$
$$\kappa = 0.33, r_1 = 1, r_2 = 2,$$
$$s_1 = [1, 0.4, 0.005, 0.1 \exp(-3), ..., 0.1 \exp(-K+1)],$$
$$s_2 = [1.3, 0.2, 0.005, 0.1 \exp(-3), ..., 0.1 \exp(-K+1)],$$
$$p_{\text{tr}} = 0.8, \epsilon = 0.5.$$

The remaining parameters are treatment effect $\mu$ and observation rate $\rho$, which we allow to vary across different trials.

