# OpenReview forum: "Modeling treatment events in disease progression"
_ICLR.cc/2020/Conference — Reject_

### Official Review · AnonReviewer1 · 2019-10-15
**Official Blind Review #1**

**Rating:** 1

**Review:**

This paper proposes to incorporate treatment effect in disease progression modeling for individual patients. It views treatment effect as an exogenous variable and represent that with one additional parameter in the model.

There is not enough novelty with the proposed method. It only extends Kidzinski and Hastie (2018) by adding one extra parameter that represents the average additive treatment effect. And this new model does not outperform the previous model in  Kidzinski and Hastie (2018) significantly.

The method also does not perform significantly better on real data. Given the real data  as a 1-d vector of length 6k, the method learns a 9x9 latent matrix to represent the data. And the method beats, but not significantly, the intuitive weak baseline rMean which uses the mean of the same patient’s previous visits for future prediction.






**Experience Assessment:**

I have read many papers in this area.

**Review Assessment: Checking Correctness Of Derivations And Theory:**

I did not assess the derivations or theory.

**Review Assessment: Checking Correctness Of Experiments:**

I did not assess the experiments.

**Review Assessment: Thoroughness In Paper Reading:**

I read the paper at least twice and used my best judgement in assessing the paper.

---

### Official Review · AnonReviewer2 · 2019-10-22
**Official Blind Review #2**

**Rating:** 1

**Review:**

The paper introduces Coordinatewise-Soft-Impute (CSI) algorithm that attempts to estimate the progression of a disease while taking into account the effect of a single binary treatment intervention. The authors also provide prove that CSI converges to the global solution regardless of the initialization. The paper is concluded with experimental results on synthetic  and real-world data, comparing the proposed method with contender methods functional Principal Component Analysis (fPCA) and Soft-Longitudinal-Impute (SLI).

This paper should be rejected due to the following arguments:
	- The authors claim that the “existing literature talks little about modeling treatment effect on disease progression.” However, this is not true. Here are a few works and the references within, for instance:
		Xu, Y., Xu, Y., & Saria, S. (2016, December). A Bayesian nonparametric approach for estimating individualized treatment-response curves. In Machine Learning for Healthcare Conference (pp. 282-300).
		Lim, B. (2018). Forecasting treatment responses over time using recurrent marginal structural networks. In Advances in Neural Information Processing Systems (pp. 7483-7493).

	- The proposed method is simple extension of that of (Kidzinski & Hastie, 2018), by adding a term that accounts for treatment and its timing. It does not present enough novelty.

	- CSI is empirically compared with fPCA and SLI, however, neither of these two methods take into account the effect of applying treatment. This comparison is unfair and therefore, cannot justify the superiority of CSI.

Things to improve the paper that did not impact the score:
	- Citations throughout the paper had wrong formatting. Often \citep should have been used instead of \citet.
	- Page 1, last par., last sentence: Thus, they need to “be” modeled differently.
	- Page 2, last par., line -2: Principle >> Principal
	- Page 3, par. 2, line 4: the observation matrix Y is N \times T (not N \times K).
	- Page 4, line 2 of Eq. (3.2): first argument misses Y.
	- Code was included but no description or readme file was provided for how to run the experiments.

**Experience Assessment:**

I have published one or two papers in this area.

**Review Assessment: Checking Correctness Of Derivations And Theory:**

I carefully checked the derivations and theory.

**Review Assessment: Checking Correctness Of Experiments:**

I carefully checked the experiments.

**Review Assessment: Thoroughness In Paper Reading:**

I read the paper thoroughly.

---

### Official Review · AnonReviewer3 · 2019-10-28
**Official Blind Review #3**

**Rating:** 1

**Review:**

This paper proposed a framework for modeling disease progression taking into account the treatment events.

Comments:
1. There is no significant theoretical innovation in this paper. Specifically, the learning approach for optimizing the parameter set W in the presence of \mu is not much different from the one proposed by (Kidzinski & Hastie, 2018) and the optimal value of \mu is simply the solution for L2 loss function. Besides, the narrative of section 2 shares too much similarity with the context of Kidzinski & Hastie, 2018.

2. The practical significance of the paper is rather limited considering that the framework can only take into account one specific treatment at a time when training a model. There are usually different treatments for a certain disease and it is hard to control enough patients to all take the same treatment. Sometimes, even the same patient could take multiple treatments at the same time or along his trajectory.

3. The paper is poorly written. There are too many grammatical errors. Some sentences are hard to understand. E.g. “Treatments of various types, however, rely on human decisions and to some extent, an exogenous variable to the development of disease. Thus they need to modeled differently.” should be “Treatments of various types, however, rely on human decisions and to some extent, are exogenous variables to the development of disease. Thus they need to be modeled differently.”. “we add a penalty term on the nuclear norm of W .”, Do you mean “we add a penalty term which is the nuclear norm of W .”? Cause I don’t see any additional penalty term in the nuclear norm compared with equation 2.2. “...where (I_S)_{i,j} = 1 if and only T_i ≥ s(i)...” should be “...where (I_S)_{i,j} = 1 if and only if T_i ≥ s(i)...”. “Finally we show that if the sequence {(\mu_{\lambda}^{(k)}, W_{\lambda}^{(k)} )}_k, it has to converge to a solution of (4.1).” Grammatically speaking, the if clause is not even a complete clause.

**Experience Assessment:**

I have read many papers in this area.

**Review Assessment: Checking Correctness Of Derivations And Theory:**

I assessed the sensibility of the derivations and theory.

**Review Assessment: Checking Correctness Of Experiments:**

I assessed the sensibility of the experiments.

**Review Assessment: Thoroughness In Paper Reading:**

I read the paper at least twice and used my best judgement in assessing the paper.

---

### Decision · Program_Chairs · 2019-12-19

**Decision:**

Reject

**Comment:**

The proposed method has very weak novelty.